# Recursive MAGUS: Scalable and accurate multiple sequence alignment

**Vladimir Smirnov** *

Department of Computer Science, University of Illinois at Urbana-Champaign, Urbana, Illinois, United States of America

* smirnov3@illinois.edu

**Data Availability Statement:** MAGUS is open-source and freely available at https://github.com/vlasmirnov/MAGUS. The datasets used in this study can be downloaded from the Illinois Data Bank at https://doi.org/10.13012/B2IDB-1048258_V1.

## Abstract

Multiple sequence alignment tools struggle to keep pace with rapidly growing sequence data, as few methods can handle large datasets while maintaining alignment accuracy. We recently introduced MAGUS, a new state-of-the-art method for aligning large numbers of sequences. In this paper, we present a comprehensive set of enhancements that allow MAGUS to align vastly larger datasets with greater speed. We compare MAGUS to other leading alignment methods on datasets of up to one million sequences. Our results demonstrate the advantages of MAGUS over other alignment software in both accuracy and speed. MAGUS is freely available in open-source form at https://github.com/vlasmirnov/MAGUS.

## Author summary

Many tasks in computational biology depend on solving the problem of multiple sequence alignment (MSA), which entails arranging a set of genetic sequences so that letters with common ancestry are stacked in the same column. This is a computationally difficult problem, particularly on large datasets; current MSA software is able to accurately align up to a few thousand sequences at a time. Unfortunately, growing biological datasets are rapidly outpacing these capabilities. We present a new version of our MAGUS alignment tool, which has been massively scaled up to handle datasets of up to one million sequences, and demonstrate MAGUS's excellent performance in aligning ultra-large datasets. The MAGUS software is open-source and can be found at https://github.com/vlasmirnov/MAGUS.

This is a *PLOS Computational Biology* Software paper.

## Introduction

One of the principal problems in computational biology is multiple sequence alignment (MSA), being necessary for a wide range of downstream applications. This challenge is well-

**Funding:** This work was funded by the Ira & Debra Cohen Graduate Fellowship to VS. VS was also funded by a research assistantship with Dr. Tandy Warnow, which was funded by NSF grant ABI-1458652. The funders had no role in study design, data collection and analysis, decision to publish, or preparation of the manuscript.

**Competing interests:** The authors have declared that no competing interests exist.

studied, and a good number of strong methods have been developed [1–8]. Most of these leading methods follow the paradigm of "progressive alignment", and are able to show reasonable accuracy and speed on datasets of modest size (a few hundred to a few thousand sequences).

Unfortunately, datasets with more sequences and greater evolutionary diameters require a different approach. Accurate progressive alignment methods rely on heuristics whose run-times scale very poorly, and early mistakes are compounded over large numbers of pairwise alignments. As a consequence, a family of divide-and-conquer methods was developed to meet the demands of larger datasets [9–11].

MAGUS (Multiple Sequence Alignment using Graph Clustering) was recently introduced [12] as a new evolution of this family. MAGUS uses the GCM (Graph Clustering Merger) technique to combine an arbitrary number of subalignments, which allows MAGUS to align large numbers of sequences with highly competitive accuracy and speed. In its original form, MAGUS is able to align up to around 40,000 sequences.

In this paper, we extend MAGUS to handle datasets of much greater size, demonstrating alignments of up to one million sequences. The next section briefly explains how MAGUS operates, and presents our extensions to enable scalability. Next, we describe our experimental study and show our results, comparing MAGUS to other methods with regard to alignment accuracy and speed over ultra-large datasets. Finally, we discuss our findings and future work.

## Design and implementation

### Overview of MAGUS

MAGUS is a recently developed divide-and-conquer alignment method that inherits the basic structure of the earlier PASTA [11] algorithm: MAGUS decomposes the dataset into subsets, aligns them piecewise, and merges these subalignments together. The basic algorithm is outlined in Fig 1 and itemized below.

**Input:** a set of unaligned sequences.

1. Construct a guide tree over the unaligned sequences. (Our default way of doing this is explained below.)

2. Use the guide tree to break the dataset into subsets. This is done by "centroid edge decomposition" [11], deleting edges to break the tree into sufficiently small, balanced pieces.

3. Align each subset with MAFFT -linsi [3].

4. Construct a set of backbone alignments spanning our subsets. Each backbone is composed of equal-sized random subsets from each subalignment and aligned with MAFFT -linsi.

5. Compile the backbones into an alignment graph. Each node represents a subalignment column, and the edges are weighted by how often they are matched by the backbone alignments.

6. Cluster the alignment graph with MCL [13].

7. Order the clusters into a valid alignment. We use a heuristic search to resolve conflicts with minimal changes.

8. Output the full alignment.

Please refer to the original paper [12] for more information. Steps 5–8 comprise GCM (Graph Clustering Merger, Fig A in S1 Text), the method by which MAGUS merges subalignments and its biggest departure from previous divide-and-conquer methods. The pipeline was built to be flexible: the user can supply their own subalignments in lieu of steps 1–4, their own

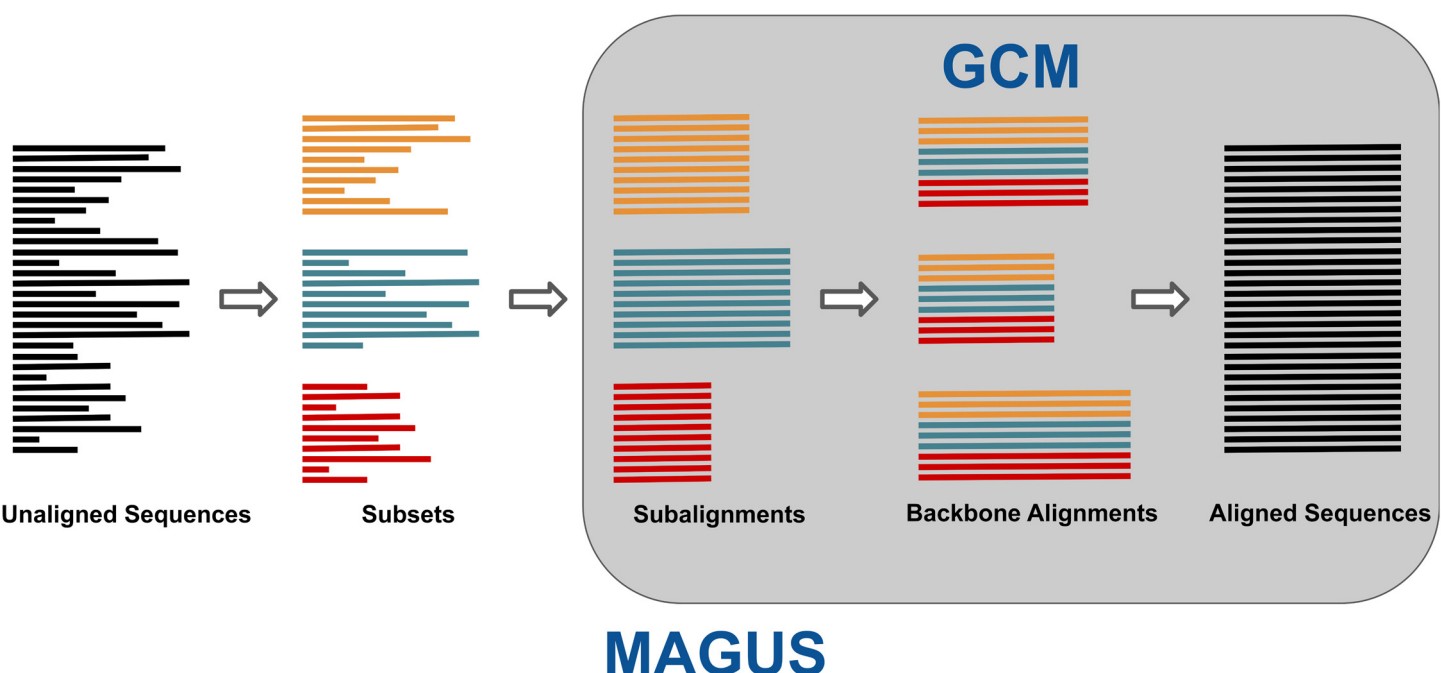

**Fig 1. MAGUS overview.** The unaligned sequences are decomposed into disjoint subsets, which are individually aligned and merged together with GCM.

guide tree for step 2, and their own backbones for step 5. The number and size of subsets and backbones can also be controlled.

## Motivation for MAGUS enhancement

Despite its advantages, the original version of MAGUS ("MAGUS 1") suffers from a number of constraints on its scalability. We motivate the need for improvement by glancing ahead to our experimental study, where MAGUS 1 is seen to struggle with increasing dataset sizes: MAGUS 1 takes over 20 hours to align 50,000 sequences and fails on larger datasets due to memory issues. In the next section, we explain the limitations of MAGUS 1 and present the improvements that comprise the paper.

## MAGUS improvements

**Recursion.** First, there is a soft limit on how many sequences MAGUS 1 can reasonably align. MAFFT -linsi [3], which is used for building subset and backbone alignments, starts to really slow down past around 200 sequences. Additionally, the cluster ordering step (step 8 above) tends to struggle with more than about 200 subsets. Therefore, assuming a practical limit of about 200 subsets of 200 sequences each, unmodified MAGUS can be expected to handle up to around $200 \times 200 = 40,000$ sequences.

We parry this limitation with a fairly straightforward recursive structure, shown in Fig 2. Instead of automatically aligning our subsets with MAFFT, subsets larger than a threshold are recursively aligned with MAGUS. This threshold can be set by the user and is, by default, the greater of the backbone size and the target subset size used for decomposition. Our subalignments are merged with GCM just as before, regardless of whether each subalignment was estimated with MAFFT or MAGUS.

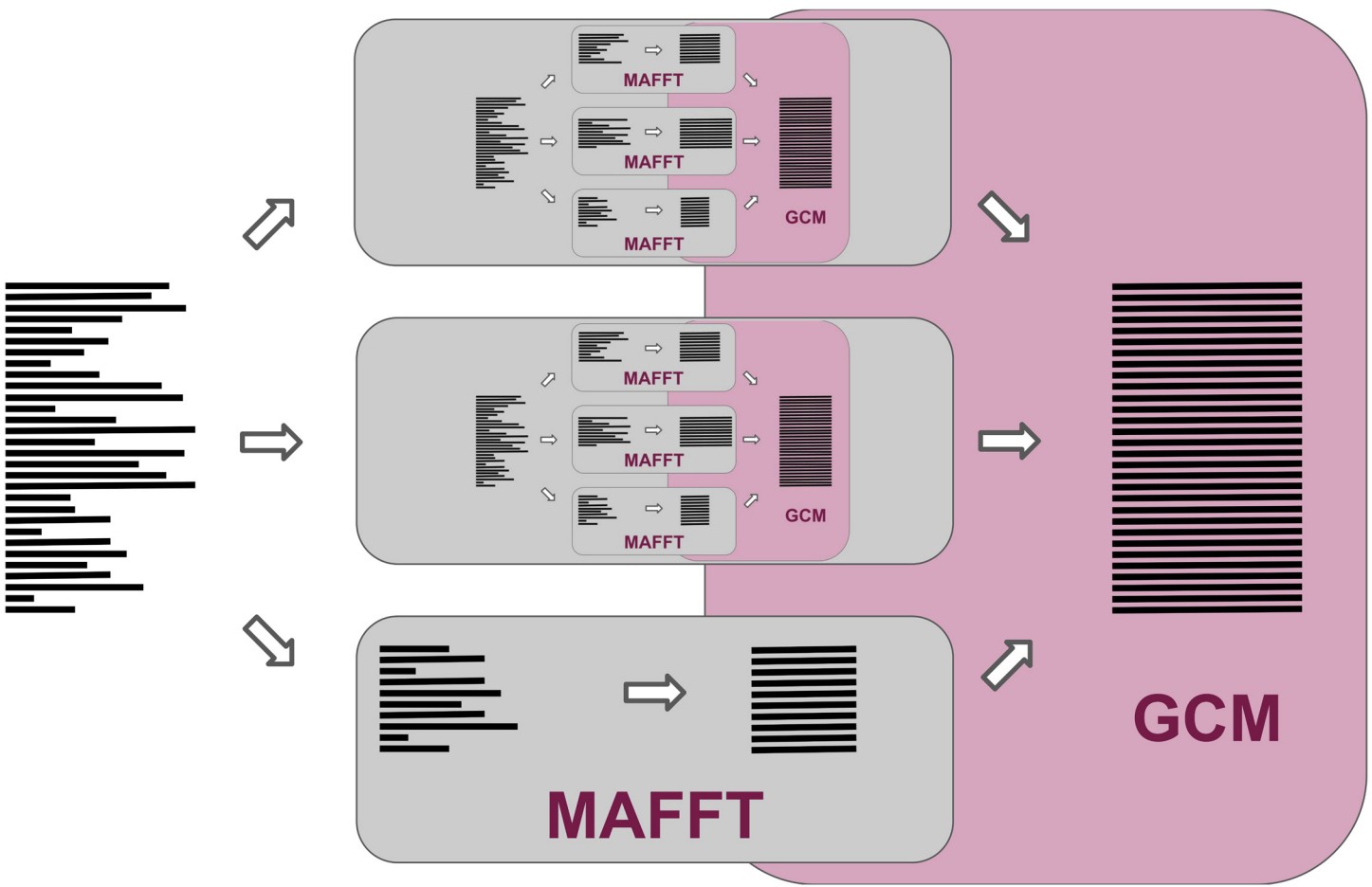

**Fig 2. Recursive MAGUS overview.** Instead of aligning all of our subsets with MAFFT, subsets larger than a given threshold are recursively aligned with MAGUS. Subsets below the threshold are aligned with MAFFT. As before, all subalignments are merged with GCM.

**Parallelism.** The next issue is parallelism. MAGUS 1 already implements thread-parallelism: it runs on a single compute node, and it can use all available threads on that node to run MAFFT tasks in parallel. This is more than enough for a few tens of thousands of sequences on a decent machine. However, with ultra-large datasets, we definitely want to benefit from node-parallelism, when multiple compute nodes can collaborate. We implement node-parallelism by extending MAGUS 1's task management code. MAGUS 1 maintains task files with MAFFT alignments and other self-contained tasks that are pending or running, which allows worker threads to divide the jobs and MAGUS to easily resume in case of failure. Reworking this system to allow for multiple compute nodes to use the same set of files effectively permits any number of nodes to join and take tasks to work on.

**Guide tree.** MAGUS decomposes the dataset into subsets by estimating a rough guide tree with FastTree [14], a fast maximum likelihood tree estimation method. Since FastTree requires an alignment, we first compile a rough alignment by aligning 300 random sequences with MAFFT and adding the remaining sequences with HMMER [15]. The guide tree is recursively broken apart until the subsets are small enough. This is the same strategy used in PASTA, and seems very difficult to improve upon. On very large alignments, however, even

FastTree becomes painfully slow (around 5 days on a million sequences, as will be shown below).

The new version of MAGUS presents a wider range of guide tree options, intended for situations where FastTree might not be fast enough or fails due to numerical issues. The guide tree can now also be generated with Clustal Omega's [2] initial tree method, MAFFT's PartTree [16] initial tree method, and FastTree's minimum evolution tree (i.e. limited to distance-based calculations without maximum likelihood). In extremis, the dataset can be decomposed randomly for maximum speed.

**Memory management and alignment compression.** Memory management becomes a salient problem when handling very large datasets. For example, without modifications, MAGUS alignments on the full million-sequence RNASim dataset fall between 1 and 3 terabytes (Fig B in S1 Text). Moreover, simply having too many subalignments loaded into memory at the same time can overrun the available RAM at such dataset sizes.

We solve the latter problem by reworking the code to ensure that at most one subalignment may be fully loaded into memory at any time. With large dataset sizes, this limits the memory complexity of MAGUS to the size of the largest subalignment.

The problem of excessively large alignments is addressed by introducing a method of conservative lossy compression. If MAGUS calculates that the size of the uncompressed alignment will exceed a threshold (100GB by default, may be set by the user), MAGUS will compress the alignment to the threshold size. The compression scheme is fairly straightforward and works by "dissolving" columns: the letters are set to lower-case and shunted to neighboring columns. If the neighboring columns already contain **lower-case** letters from the same sequences, these are also shunted away in a recursive domino effect. (If the neighboring columns already contain **upper-case** letters from the same sequences, then the move is invalid.) Columns are dissolved one at a time, starting with those containing the fewest letters, until the threshold is reached or no more valid moves remain. Please refer to Fig 3 for an example.

Note that if we "dissolve" a column with only one upper-case letter, then no homologous pairs are lost. Thus, the compression procedure remains lossless for as long as we are only dissolving such columns, and MAGUS allows the user to request lossless compression.

Table B in S1 Text shows the effect of compression on MAGUS's RNASim alignments at various sizes. At one million sequences, for example, the uncompressed alignment is about 1037GB, which can be reduced to 591GB with lossless compression, and reduced further to 25GB with lossy compression. Similarly, the uncompressed alignment over 500,000 sequences is 366GB, falling to 193GB with lossless compression and 10GB with lossy compression. Lossy compression increases the SP error by less than one millionth on these datasets, so it is generally safe to use.

## Results

### Experimental design

Our experimental design is outlined below.

The preliminary portion of our study that explores the effects of our MAGUS extensions described above, using MAGUS 1 as our baseline. We test the impact of compression on alignment error, the use of different guide trees, and the benefit of node-parallelism. Due to space limitations, these results are available in the Supplementary Materials.

Our subsequent experiments compare MAGUS against a range of competing methods across all of our datasets. This is the most important part of our study, intended to exercise the current state-of-the-art in the alignment of ultra-large nucleotide and protein datasets. We present our results below.

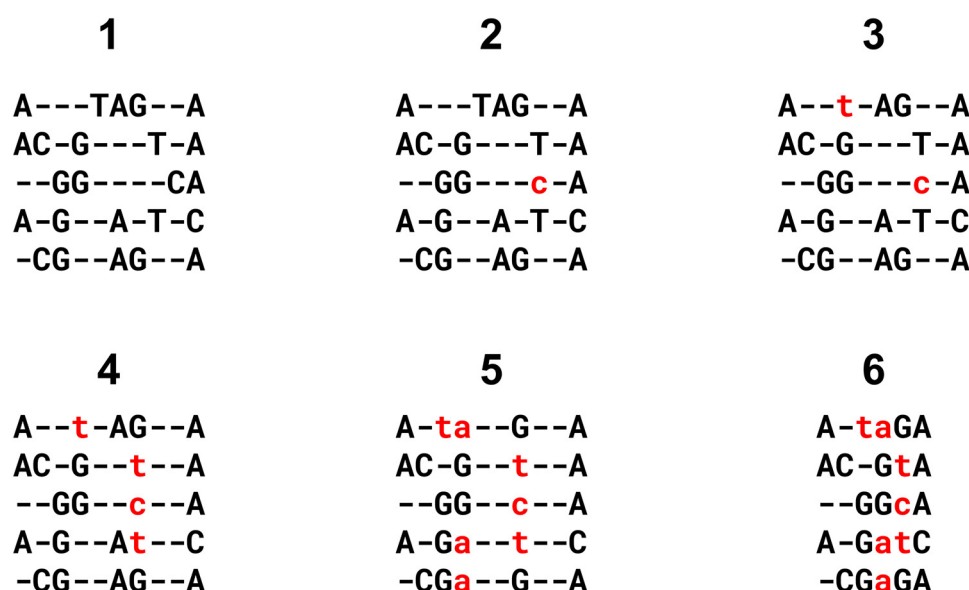

**Fig 3. Alignment compression example.** At each step, we dissolve the column with the smallest number of upper-case letters (i.e., homologous pairs), that can be merged sideways without displacing upper-case letters in another column. Dissolved letters become lower-case and no longer represent homology. Steps 2 and 3 dissolve singleton columns, and are thus lossless. Steps 4 and 5 are lossy. Note step 5, where the lowercase 't' in the destination column was shunted further left to make room. Step 6 simply disposes of the empty columns to form the final, compressed alignment.

**Datasets.** Our study uses a number of simulated and biological datasets from previous publications [4, 11]. Please see Table 1 for dataset statistics. These datasets were selected to provide suitably large and varied alignment problems with reference alignments, containing both nucleotide and amino acid sequences.

- **RNASim:** [11] This is a simulated RNA dataset, generated under a non-homogeneous model of evolution that does not conform to the usual GTR model assumptions. We use sub-samples ranging from 10,000 to the full one million sequences, with one replicate per size.

- **16S:** [17] We use three large biological nucleotide datasets from the Comparative Ribosomal Website: 16S.3, 16S.T, and 16S.B.ALL, with 6,323, 7,350, and 27,643 sequences, respectively.

- **HomFam:** [2] Finally, we include 19 amino acid HomFam datasets from, which have small Homstrad reference alignments on 5–20 sequences each. These datasets range from 10,099 to 93,681 sequences and allow us to evaluate our methods on large protein datasets. (Following the PASTA paper, we exclude the "rhv" dataset due to having a weak alignment).

**Methods.** We compare the following methods in our study, taken from previous publications [4, 11]. To the best of our knowledge, these methods are presently the best-equipped to tackle very large multiple sequence alignments. Regressive T-Coffee [18] is another recent development, but we were unable to run it on Blue Waters.

- **MAGUS 1** We use the original MAGUS as a baseline. This version does not use recursion or compression, uses a FastTree decomposition, and can only run on a single node.

- **MAGUS** The latest version takes advantage of the new features detailed above. We enable recursion and compress alignments above 100GB. In addition to the default FastTree

**Table 1. Dataset properties.** Statistics taken from [11]. P-distance denotes the normalized Hamming distance, or the fraction of non-gap letter pairs that do not match. Alignment length shows the length of the reference alignment.

| Dataset | # Seqs | Avg. p-dist. | Max p-dist. | % gaps | align. length | type |
|---|---|---|---|---|---|---|
| RNASim | 10,000–1,000,000 | 0.41 | 0.61 | 93 | 18,268 | sim NT |
| 16S | | | | | | |
| - 16S.3 | 6,323 | 0.32 | 0.83 | 82 | 8,716 | bio NT |
| - 16S.T | 7,350 | 0.35 | 0.90 | 87 | 11,856 | bio NT |
| - 16S.B.ALL | 27,643 | 0.21 | 0.77 | 80 | 6,857 | bio NT |
| HomFam | | | | | | |
| - gluts | 10,099 | 0.60 | 0.81 | 8 | 235 | bio AA |
| - myb-DNA-binding | 10,398 | 0.59 | 0.77 | 12 | 61 | bio AA |
| - tRNA-synt-2b | 11,293 | 0.81 | 0.88 | 34 | 467 | bio AA |
| - biotin-lipoyl | 11,833 | 0.71 | 0.84 | 26 | 112 | bio AA |
| - hom | 12,037 | 0.64 | 0.84 | 35 | 98 | bio AA |
| - ghf13 | 12,607 | 0.72 | 0.84 | 25 | 626 | bio AA |
| - aldosered | 13,277 | 0.57 | 0.79 | 19 | 386 | bio AA |
| - hla | 13,465 | 0.24 | 0.33 | 0 | 178 | bio AA |
| - Rhodanese | 14,049 | 0.76 | 0.89 | 31 | 216 | bio AA |
| - PDZ | 14,950 | 0.69 | 0.84 | 15 | 110 | bio AA |
| - blmb | 17,200 | 0.79 | 0.90 | 30 | 344 | bio AA |
| - p450 | 21,013 | 0.79 | 0.87 | 20 | 512 | bio AA |
| - adh | 21,331 | 0.36 | 0.47 | 0 | 375 | bio AA |
| - aat | 25,100 | 0.71 | 0.87 | 15 | 476 | bio AA |
| - rrm | 27,610 | 0.77 | 0.91 | 45 | 157 | bio AA |
| - Acetyltransf | 46,285 | 0.75 | 0.87 | 29 | 229 | bio AA |
| - sdr | 50,157 | 0.77 | 0.89 | 28 | 361 | bio AA |
| - zf-CCHH | 88,345 | 0.65 | 0.85 | 25 | 39 | bio AA |
| - rvp | 93,681 | 0.63 | 0.76 | 19 | 132 | bio AA |

decomposition, we explore other guide trees: FastTree without Maximum Likelihood, MAFFT's PartTree, Clustal Omega's initial tree method, and a random decomposition. Henceforth, we indicate the guide tree and use of recursion in parentheses. For example, MAGUS(Recurse, Clustal) denotes MAGUS using Clustal Omega's guide tree and with recursion enabled.

- **PASTA** [11]

- **UPP** [4]

- **UPP(Fast)** We use the "Fast" mode described in the UPP paper.

- **Muscle** [1]

- **Clustal Omega** [2]

- **MAFFT -auto** [3]. The "auto" mode directs MAFFT to choose an appropriate alignment strategy based on the input dataset.

**Error metrics.** We evaluate alignment accuracy using SPFP/SPFN (Sum-of-Pairs False Positives and Negatives) rates, computed using FastSP [19]. These values represent the fractions of missing and incorrect homologous pairs in the estimated alignment. For convenience,

we show the average of SPFP and SPFN as a single "SP error" in the main paper; SPFP and SPFN are shown separately in the Supplementary Materials. Our estimated alignments are compared against the true alignment on RNASim and the curated reference alignments on 16S. The HomFam datasets provide reference alignments over a small number of included sequences; we compute our alignment error over just these reference sequences.

**Computing resources.**   We used the NCSA Blue Waters supercomputer for our experiments. Our jobs were run on nodes with 32 cores, 64GB of RAM, and a maximum wall time of 7 days.

## Experimental results

The preliminary part of our study, which investigates the impacts of compression, guide tree selection, and node-parallelism, is available in the Supplementary Materials (due to space constraints). These results provide us with two natural guide tree choices for MAGUS: using FastTree (the default, described above) is the most accurate, while using Clustal Omega's initial tree is the faster alternative. Here, we present the principal part of our study, where we compare MAGUS to our other methods across all of our datasets.

**HomFam.**   Our first set of results concern the HomFam protein datasets. The error rates are averaged in Fig 4, and the complete results for all datasets are available in Table C in S1 Text. These results show more variability than the other datasets, but the general trends are as follows. Muscle and Clustal trail the others, averaging 46.6% and 27.2% error, respectively. MAFFT, UPP, and PASTA are all on par, averaging about 21–23% error. The MAGUS versions perform markedly better: MAGUS(Recurse, Clustal) yields 17.9% error, MAGUS (Recurse, FastTree) shows 16.5%, and MAGUS 1 leads with 15.5%. Furthermore, MAGUS 1 achieves the best result on 12 of the 19 datasets. Recursive MAGUS (both versions) accounts for 2 of the others, while Clustal and UPP each do best on 2.

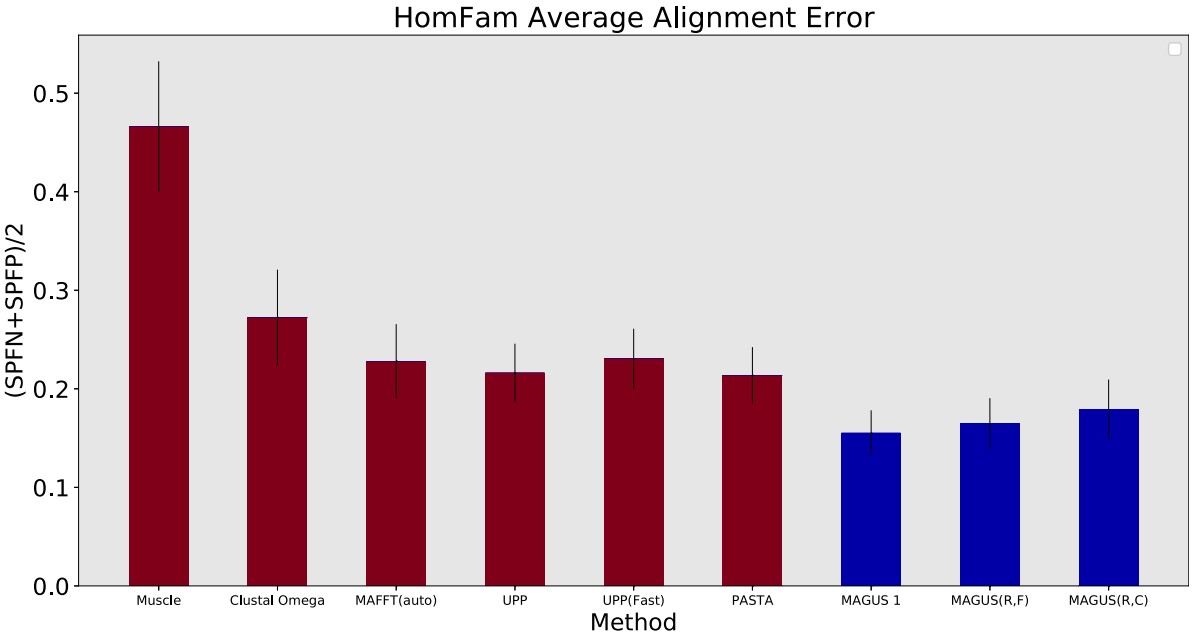

**Fig 4. Average SP error on HomFam datasets.** Error is the average of SPFP and SPFN. Results are averaged over the datasets where all methods completed (Muscle segfaulted on two). Error bars show standard error. MAGUS was run with the default 25 subsets.

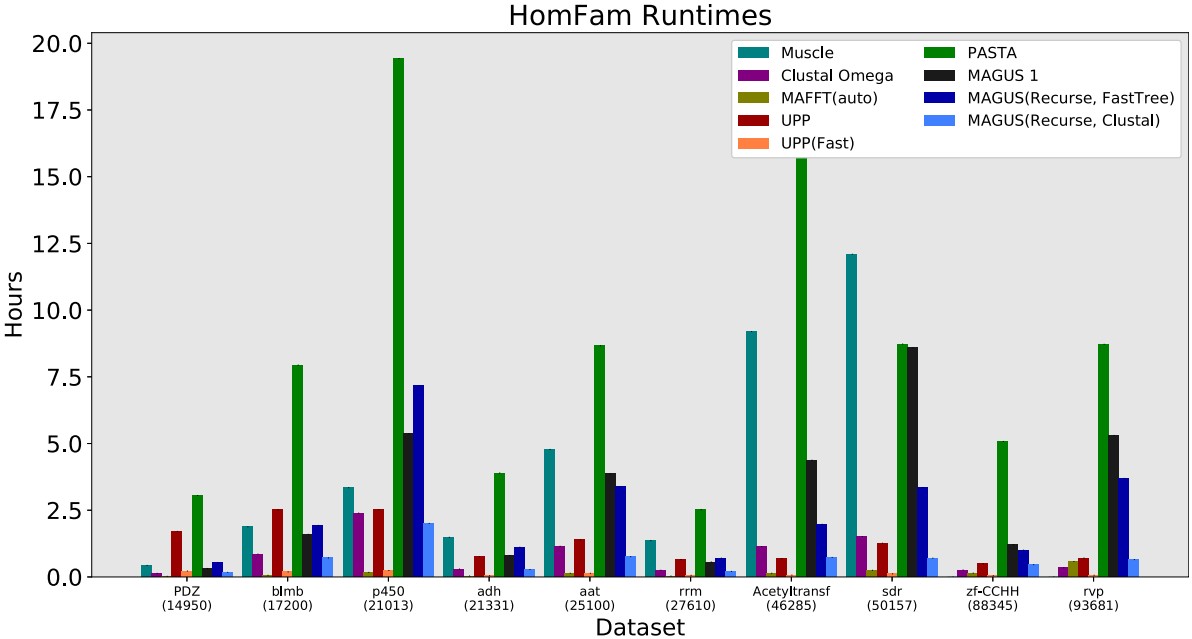

**Fig 5. Homfam (largest 10 datasets) runtime, all methods.** MAGUS was run with the default 25 subsets. Muscle segfaulted on the two largest datasets.

The HomFam runtime results are shown in Fig 5 and Fig N in S1 Text. PASTA is visibly the slowest, taking about 2–5 hours on the smaller datasets and up to 20 hours on the larger ones. MAFFT, UPP(Fast), Clustal Omega, and MAGUS(Recurse, Clustal) are the fastest, generally finishing in a few minutes to an hour. Notably, we see MAGUS 1 begin to dramatically slow down without recursion, running longer than MAGUS(Recurse, FastTree) on the largest datasets.

**16S.** The next set of results pertain to the biological 16S datasets, shown in Figs 6 and 7. As above, Muscle and Clustal trail the other methods in accuracy. On the smallest dataset, 16S.3, the results are fairly close: UPP(Fast), PASTA, and all versions of MAGUS are at about 19% SP error. There is a larger difference on 16S.T, with PASTA at around 23%, UPP and UPP(Fast) around 21%, and all versions of MAGUS at about 20%. Lastly, UPP, PASTA, and MAGUS are again fairly close on 16S.B.ALL; PASTA shows about 11% error, while both versions of UPP and MAGUS have about 10.5% error.

In terms of runtime, we see that UPP, PASTA, and both versions of recursive MAGUS are the slowest methods on 16S.3 and 16S.T, running around 4–5 hours. The fastest method is MAFFT(auto) at about 2 minutes, while Muscle and UPP(Fast) take about half an hour. The picture is a little different on 16S.B.ALL, where Muscle, UPP, and PASTA seem to drastically slow down; they take about 11, 14, and 17 hours, respectively. MAGUS 1 also falters here, taking 18 hours, while recursive MAGUS with FastTree and Clustal only increases to 8 and 4 hours, respectively. MAFFT and UPP(Fast) remain the fastest, only taking 1–2 hours.

**RNASim.** In the final part of our study, we probe the limits of scalability on the RNASim datasets. Figs 8 and 9 show us the error and runtime results, while Table 2 summarizes all method failures. Muscle is the worst performer here, with 65–70% error and segfaulting after 50,000 sequences. Clustal Omega does better, with errors between about 30% and 60%, running out of time after 200,000 sequences. Then comes MAFFT -auto, with a steady error of

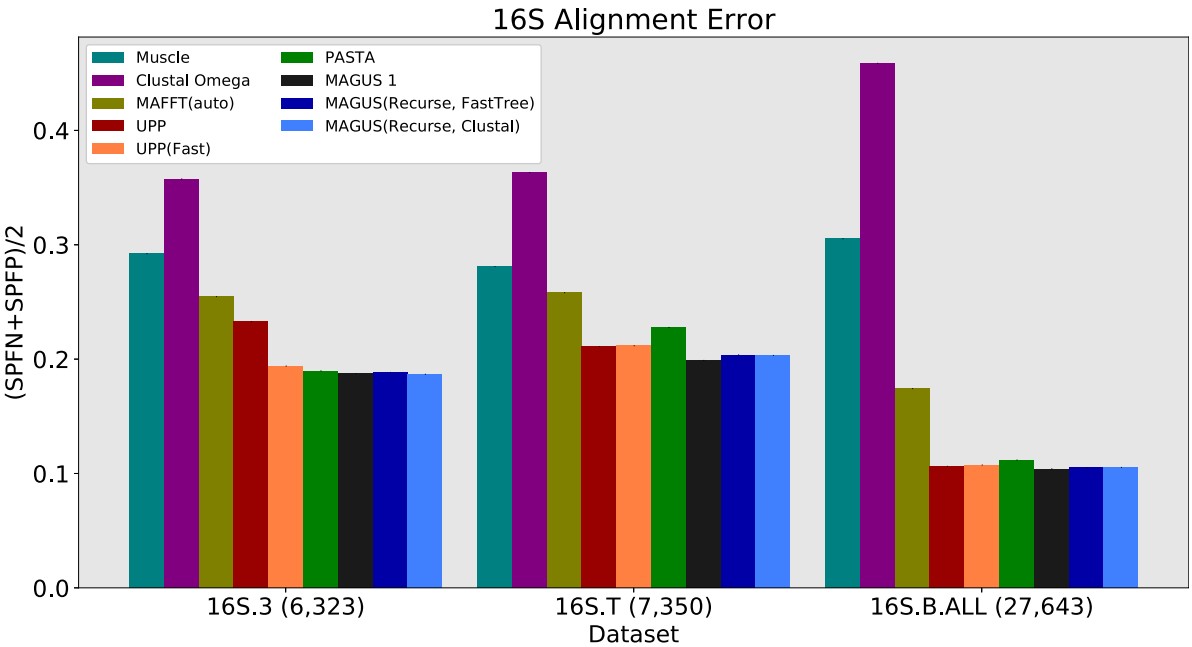

**Fig 6. 16S alignment error, all methods.** Error is the average of SPFP and SPFN. MAGUS was run with the default 25 subsets.

25–30% up to 100,000 sequences. Oddly, even though it is one of the fastest methods at 100,000 sequences (about 3.6 hours), it runs out of time at 200,000 sequences.

The accuracy of our remaining methods is shown more clearly in Fig 10. UPP(Fast) trails the other methods in accuracy, with about 2% higher error than PASTA and UPP. PASTA and UPP are about the same at around 10% error. MAGUS 1 and recursive MAGUS (both

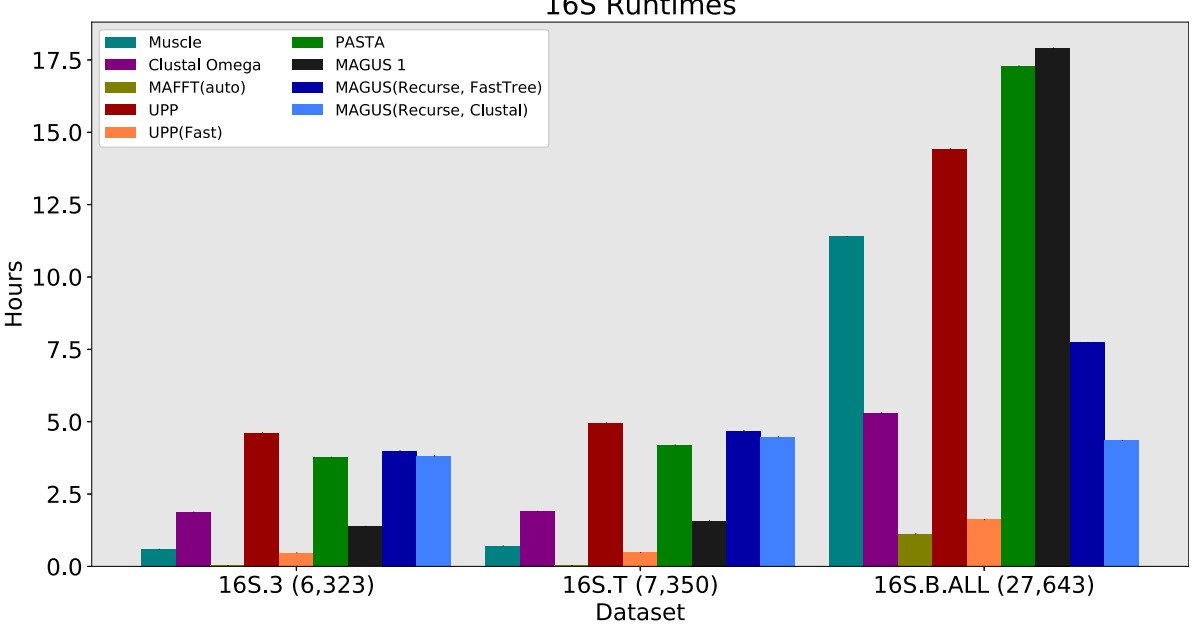

**Fig 7. 16S runtime, all methods.** Runtime is shown in hours. MAGUS was run with the default 25 subsets. MAFFT -auto completed in a few minutes on 16S.3 and 16S.T.

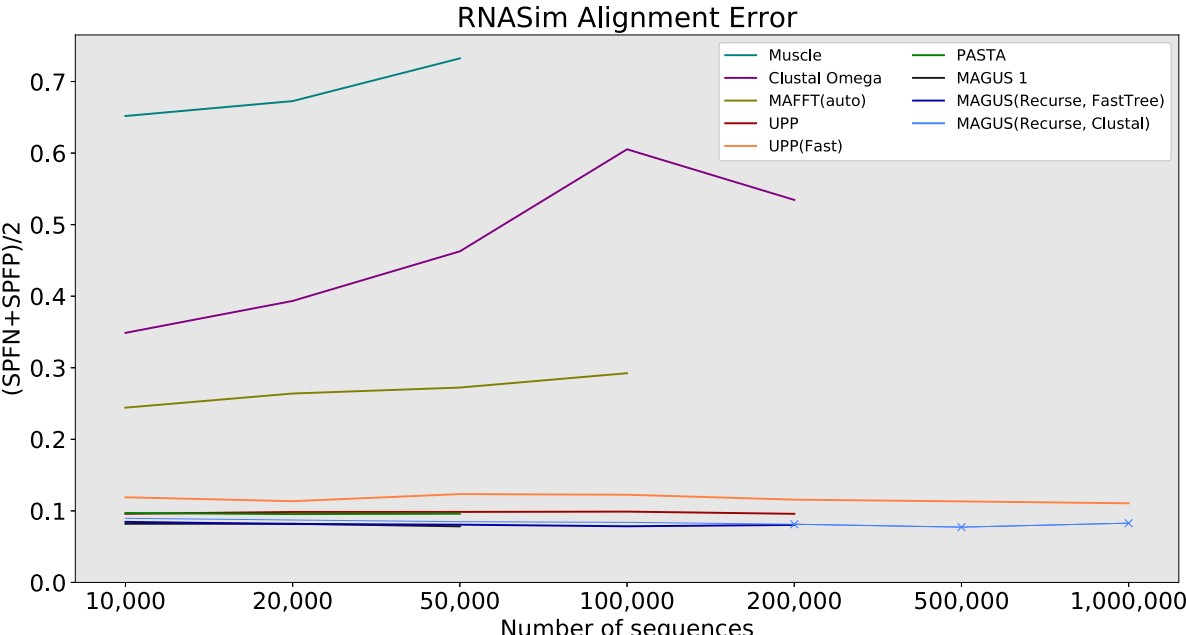

**Fig 8. RNASim alignment error, all methods.** Error is the average of SPFP and SPFN. 'X' markers indicate that compression was used (MAGUS alignments above 100GB). MAGUS was run with 100 subsets on RNASim to reduce load on Blue Waters. Compute nodes had 64GB of RAM and a maximum wall time of 7 days.

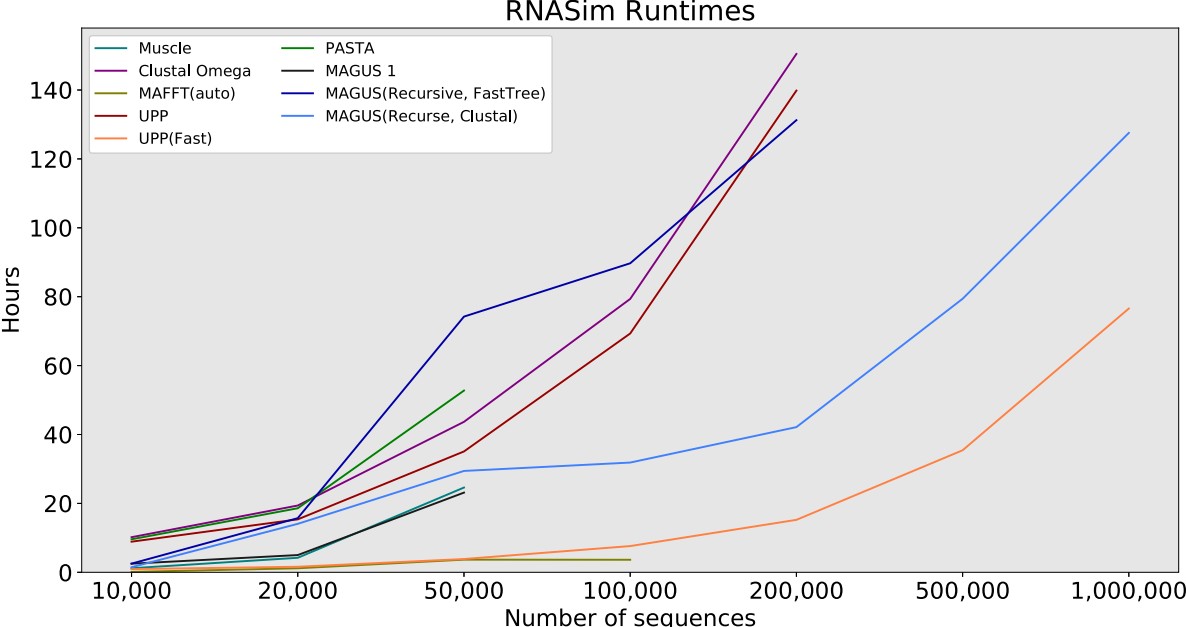

**Fig 9. RNASim runtime, all methods.** Runtime is shown in hours. MAGUS was run with 100 subsets on RNASim to reduce load on Blue Waters. Compute nodes had 64GB of RAM and a maximum wall time of 7 days.

**Table 2. Method failures on RNASim.** PASTA and MAGUS 1 failed due to excessive memory usage; compute nodes had 64GB of memory.

| Method | Highest # Aligned | Failure |
|---|---|---|
| Muscle | 50,000 | "segmentation fault" |
| Clustal Omega | 200,000 | Max runtime elapsed (7 days) |
| MAFFT(auto) | 100,000 | Max runtime elapsed (7 days) |
| UPP | 200,000 | Max runtime elapsed (7 days) |
| PASTA | 50,000 | "Error detected during page fault processing. Process terminated via bus error." |
| MAGUS 1 | 50,000 | "OOM killer terminated this process." |

versions) have the best accuracy. MAGUS 1 is the most accurate at 10,000–50,000 sequences (8.2–7.8% error), but can't proceed beyond that. MAGUS(Recurse, FastTree) is second-best at about 8.5–8%. MAGUS(Recurse, Clustal) consistently trails MAGUS(Recurse, FastTree) by about 0.5% below 200,000 sequences, and declines to about 8.3% on 1,000,000 sequences.

Aside from MAGUS(Recurse, Clustal), UPP(Fast) is the only other method that aligned all 1,000,000 sequences in a week; UPP took about 77 hours to align all 1,000,000 sequences, while MAGUS(Recurse, Clustal) took about 128 hours. PASTA encountered memory issues, while UPP and MAGUS(Recurse, FastTree) ran out of time. Notably, UPP, Clustal Omega, and MAGUS(Recurse, FastTree) showed comparable runtime scaling, all three just meeting the 1 week time limit at 200,000 sequences. MAGUS 1 initially scales better than recursive MAGUS on a single node, but only reaches 50,000.

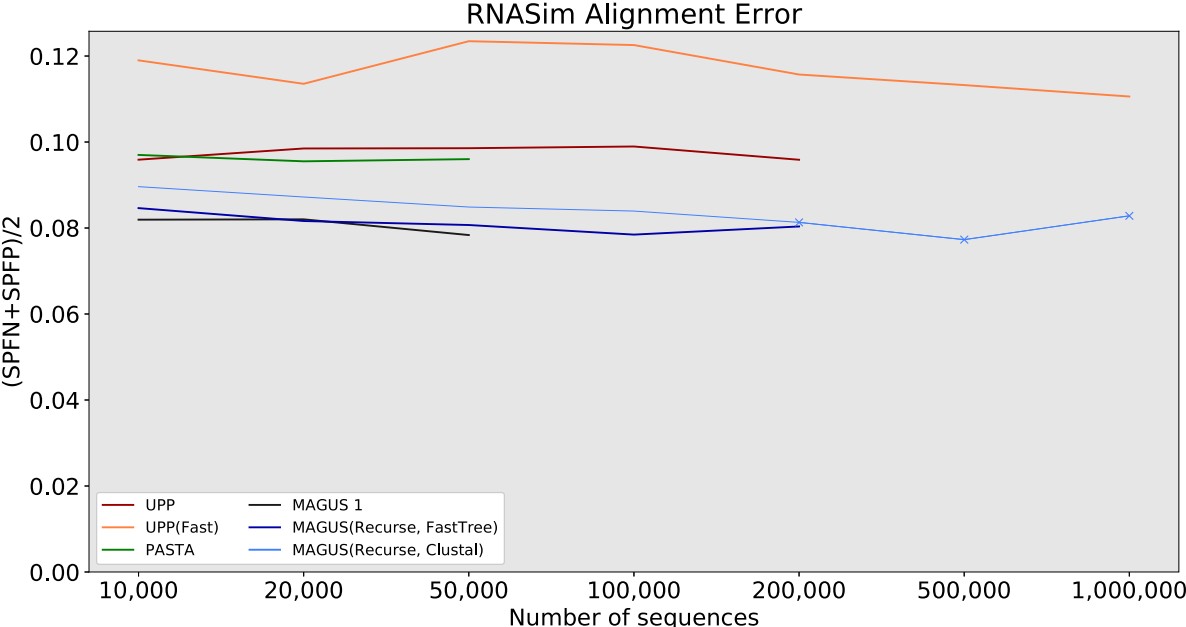

**Fig 10. RNASim alignment error, best methods.** Error is the average of SPFP and SPFN. 'X' markers indicate that compression was used (MAGUS alignments above 100GB). MAGUS was run with 100 subsets on RNASim to reduce load on Blue Waters. Compute nodes had 64GB of RAM and a maximum wall time of 7 days.

## Discussion

The accuracy of MAGUS convincingly exceeds the other methods we tried on the datasets in our study. As shown in Figs 4, 6 and 8, this is true regardless of whether recursion is used, and whether FastTree or Clustal is used for decomposition. The more difficult question we need to tease apart concerns the different ways of running MAGUS, and how they affect scalability and accuracy. We do this by considering recursion, guide tree, and node-parallelism in turn.

On one hand, recursion actually slows MAGUS down on smaller datasets. On the other hand, this is rapidly reversed as MAGUS chokes on larger datasets without recursion. This can be seen from our 16S results, where MAGUS is much faster without recursion on 6,000–7,000 sequences, but much slower on 27,000. This reversal can also be seen on the HomFam datasets. On RNASim, MAGUS without recursion is faster on 10,000–50,000 sequences, but simply fails after that.

The nature of this limitation is fairly clear: given $N$ sequences and $S$ subsets, MAGUS without recursion must run MAFFT -linsi on chunks of $\frac{N}{S}$ sequences. Thus, MAGUS without recursion is only viable for as long as MAFFT -linsi can handle these chunks. Our results suggest that subsets approaching around 1,000 sequences really become a problem: this is about where RNASim fails and 16S.B.ALL takes an inordinate amount of time. There is less of a problem on HomFam, where the amino acid sequences are much shorter.

Moreover, recursion does not improve accuracy; MAGUS without recursion is noticeably more accurate on HomFam, about the same on 16S, and slightly better on 10,000 sequences of RNASim. These observations suggest that recursion should be avoided if possible, and only engaged when the dataset becomes too large for the subsets to be reasonably aligned with the base method.

As far as decomposition strategy is concerned, the FastTree method remains the most accurate. The runtime becomes an issue on the largest datasets, where the tree takes about 5 days to compute on 1 million sequences. The best alternative, as suggested by our results, is to use the Clustal Omega guide tree. This gives the best compromise between accuracy and runtime, and only takes 14 hours on 1 million sequences.

Taking advantage of our newfound node-parallelism has a considerable impact on runtime. If we exclude the FastTree computation from the MAGUS runtime on 1 million sequences, the actual alignment stage takes about 9 days on a single node, but only about 17 hours on 10 nodes and 2.5 hours on 100 nodes. Thus, given enough compute nodes, the total runtime is mostly dominated by the guide tree method, rather than the alignment itself; this is the motivation for considering Clustal as a FastTree alternative.

## Conclusions

We presented a powerful set of improvements to our MAGUS method, allowing it to scale from 50,000 to a full million sequences. Moreover, MAGUS is able to align such vast datasets more accurately than the other methods we compared against.

UPP(Fast) remains the fastest way to effectively align a million sequences on a single compute node, but suffers from consistently worse alignment accuracy. Other methods are able to finish quickly on smaller datasets, but struggle to complete on larger numbers of sequences, while also trailing MAGUS in accuracy.

We conclude by distilling our results into a number of concrete recommendations for interested practitioners.

**Recursion is harmful on smaller datasets, but necessary on larger datasets.** If the dataset is small enough, MAGUS will run considerably faster without recursion and might have slightly better accuracy. On larger datasets, MAGUS will rapidly grind to a halt without

recursion. Thus, it is advised to avoid recursion if the dataset permits this. This threshold is dictated by subset size ($\frac{\#\ sequences}{subsets}$). Given our data, we found the "threshold" subset size to be somewhere around 1,000 sequences of a few thousand nucleotides, or somewhere above 4,000 sequences of a few hundred amino acids.

**The importance of node-parallelism and guide tree.** The default FastTree-based subset decomposition gives the best accuracy, and is fast enough for most purposes. For huge datasets of half a million or more, the Clustal Omega-based decomposition runs much faster and is nearly as accurate. As one might expect, using as many compute nodes as possible will improve the runtime. However, using more nodes than subsets will decrease the added gains from node-parallelism.

**Running MAGUS.** Putting all of the above together, the most accurate way of running MAGUS is to use the default FastTree-based decomposition without recursion, preferably on as many compute nodes as are available. If the dataset is too large to allow the subsets to align in a reasonable amount of time, recursion should be enabled. Finally, if the dataset is too large to allow FastTree to finish in a reasonable amount of time, the Clustal-based decomposition should be used.

## Future directions

We plan to explore several future directions towards further improving MAGUS. The first is to comprehensively investigate the performance of MAGUS on fragmentary data. Fragmentary sequences can potentially confound effective methods, and we will extend MAGUS to reliably handle such scenarios.

The second avenue of improvement is to consider alternative procedures for assembling backbone alignments, and is intended to further increase alignment accuracy. Currently, MAGUS uses the simple expedient of building backbones with equal, random samples from each subset. We will develop and evaluate ways to build more compact (and, thus, more accurate) backbone sets that still sufficiently span the subsets.

Thirdly, we have mostly developed MAGUS to be able to align vast numbers of sequences accurately. In the future, we hope to also extend MAGUS "in the other direction"—to handle datasets with arbitrarily long, even genome-scale sequences.

A final issue to explore is the utility and management of extra-large alignments for downstream applications. In the context of large-scale tree estimation in particular, is it better to compile a single MSA (probably with some necessary loss to compression) and use it to estimate the entire tree in one operation, or would it be more effective to estimate smaller alignments and use them for piecewise tree estimation? There has been some recent work comparing unitary and piecewise maximum likelihood tree estimation strategies on large datasets [20], showing that divide-and-conquer methods are much faster and nearly as accurate, but more investigation will be needed—particularly at the higher scales we explored here.

## Commands used

**MAGUS**
```
python3 magus.py -d tempdir -o result.txt -i unalign.txt
-t <guide tree option or path> -recurse <true|false>
-maxnumsubsets <25|100>
```

**PASTA 1.8.3**
```
python3 run\_pasta.py -i unalign.txt -o result.txt
-temporaries tempdir -d <dna|rna|protein> -keeptemp
```

**UPP 4.3.10**
```
python3 run\_upp.py -s unalign.txt -p result.txt -m rna
```

**UPP(Fast) 4.3.10**
```
python3 run\_upp.py -s unalign.txt -p result.txt -B 100 -m rna
```

**Muscle 3.8.425**
```
muscle -maxiters 2 -in unalign.txt -out result.txt
```

**Clustal Omega 1.2.4**
```
clustalo -i unalign.txt -o result.txt -threads = 32
```

**MAFFT 7.450 –auto**
```
mafft -auto -ep 0.123 -quiet -thread 32 -anysymbol
unalign.txt > result.txt
```

**FastSP 1.6.0 (Computing alignment error)**
```
java -Xmx256G -jar FastSP\_1.6.0.jar -r reference\_align.txt
-e estimated\_align.txt -ml
```

**MAGUS**
```
python3 magus.py -d tempdir -o result.txt -i unalign.txt
-t <guide tree option or path> -recurse <true|false>
-maxnumsubsets <25|100>
```

**PASTA 1.8.3**
```
python3 run\_pasta.py -i unalign.txt -o result.txt
-temporaries tempdir -d <dna|rna|protein> -keeptemp
```

**UPP 4.3.10**
```
python3 run\_upp.py -s unalign.txt -p result.txt -m rna
```

**UPP(Fast) 4.3.10**
```
python3 run\_upp.py -s unalign.txt -p result.txt -B 100 -m
rna
```

**Muscle 3.8.425**
```
muscle -maxiters 2 -in unalign.txt -out result.txt
```

**Clustal Omega 1.2.4**
```
clustalo -i unalign.txt -o result.txt -threads = 32
```

**MAFFT 7.450 –auto**
```
mafft -auto -ep 0.123 -quiet -thread 32 -anysymbol
unalign.txt > result.txt
```

**FastSP 1.6.0 (Computing alignment error)**

```
java -Xmx256G -jar FastSP\_1.6.0.jar -r reference\_align.txt
-e estimated\_align.txt -ml
```

## Supporting information

**S1 Text. Supplementary materials.** Fig A. GCM overview. Fig B. RNASim alignment sizes, MAGUS variants only. Fig C. RNASim alignment error, MAGUS variants only. Fig D. RNA-Sim runtimes, MAGUS guide trees only. Fig E. RNASim runtimes, MAGUS variants only. Fig F. RNASim SPFN error, MAGUS variants only. Fig G. RNASim SPFP error, MAGUS variants only. Fig H. RNASim SPFN error. Fig I. RNASim SPFP error. Fig J. 16S SPFN error. Fig K. 16S SPFP error. Fig L. HomFam (smallest 9 datasets) alignment error. Fig M. Homfam (largest 10 datasets) alignment error. Fig N. Homfam (smallest 9 datasets) runtime. Table A. RNASim log-scale alignment sizes. Table B. RNASim Delta error from lossy compression. Table C. HomFam (all datasets) alignment error. Table D. HomFam (all datasets) SPFN error. Table E. HomFam (all datasets) SPFP error.
(PDF)

## Author Contributions

**Conceptualization:** Vladimir Smirnov.

**Data curation:** Vladimir Smirnov.

**Formal analysis:** Vladimir Smirnov.

**Funding acquisition:** Vladimir Smirnov.

**Investigation:** Vladimir Smirnov.

**Methodology:** Vladimir Smirnov.

**Project administration:** Vladimir Smirnov.

**Resources:** Vladimir Smirnov.

**Software:** Vladimir Smirnov.

**Supervision:** Vladimir Smirnov.

**Validation:** Vladimir Smirnov.

**Visualization:** Vladimir Smirnov.

**Writing – original draft:** Vladimir Smirnov.

**Writing – review & editing:** Vladimir Smirnov.

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
