## [Decision Letter · Decision Letter 0]

12 Aug 2021

Dear Mr. Smirnov,

Thank you very much for submitting your manuscript "Recursive MAGUS: scalable and accurate multiple sequence alignment" for consideration at PLOS Computational Biology.

As with all papers reviewed by the journal, your manuscript was reviewed by members of the editorial board and by several independent reviewers. In light of the reviews (below this email), we would like to invite the resubmission of a significantly-revised version that takes into account the reviewers' comments.

We cannot make any decision about publication until we have seen the revised manuscript and your response to the reviewers' comments. Your revised manuscript is also likely to be sent to reviewers for further evaluation.

Sincerely,

Dina Schneidman-Duhovny

Software Editor

PLOS Computational Biology

Reviewer's Responses to Questions

**Comments to the Authors:**

Reviewer #1: The author presents an update to a recently-published tool (MAGUS; Smirnov & Warnow, 2020). For context, MAGUS is a divide-and-conquer approach for Multiple Sequence Alignment (MSA) that is similar in spirit to PASTA (Mirarab et al., 2015) in that a complete sequence dataset is decomposed into smaller subsets, and the subsets are aligned and merged into a single MSA using a guide tree. The novelty of MAGUS (from the original manuscript) is the use of a novel "Graph Clustering Merger" approach for merging the subset alignments.

In this manuscript specifically, the author expands upon MAGUS to improve scalability in 4 key ways:

1. Recursion: To better handle ultra-large datasets in which even the subsets are prohibitively large for the underlying aligner (e.g. MAFFT), MAGUS can now recursively call itself on the subsets (i.e., to break them down into *even smaller* subsets). This is a clever idea, and I'm excited to see that it yielded significant speed-up with respect to the original MAGUS approach, but the notion of breaking down a divide-and-conquer algorithm into smaller versions of itself recursively is fairly standard (I would even argue that *all* divide-and-conquer algorithms are inherently recursive, and in this context, the only theoretical change is that the MAGUS "base case" of using MAFFT is being made smaller).

2. Parallelism: Prior to this manuscript, MAGUS supported thread-parallelism (i.e., it could utilize all available threads on a single node), and in this update, MAGUS now also supports node-parallelism (i.e., it can now distribute tasks to multiple compute nodes, with each supporting thread-parallelism). This is an excellent additional feature that I'm excited to see implemented, but the notion of sending the individual components of a divide-and-conquer algorithm to multiple compute nodes is also fairly standard from a technical standpoint.

3. Guide Tree: In the original manuscript, MAGUS used FastTree to estimate a rough guide tree (via the maximum-likelihood approach) with which to decompose the dataset. In this update, MAGUS now supports more types of trees (e.g. to enable using a faster tree-construction approach at the expense of accuracy). This is a nice addition as far as the software itself is concerned, but from a theoretical standpoint, this is a fairly trivial update (the fundamental approach is still the same, just with the newly-added ability to swap in different tools to construct this initial guide tree).

4. Memory Management and Alignment Compression: To reduce the memory complexity of MAGUS, the author has implemented optimizations on two fronts. First, memory management is conducted more optimally by only fully loading a single subalignment into memory at any given time. This is a nice fix that I'm very excited to see implemented into MAGUS, but this is certainly more of a code revision rather than a novel approach: only loading pieces of a dataset into memory at any given time has been a standard systems programming approach for decades. Second, to *further* improve the memory complexity of MAGUS, the author has now also implemented a lossy compression scheme to "dissolve" neighboring columns that are highly-similar. I actually found the idea quite interesting, and more exploration of how this lossy compression scheme impacts accuracy would have been nice to see, especially as a function of the threshold.

The results are nice to see, but they are unsurprising: nice (but not inherently novel) optimizations were made to the MAGUS codebase, and the MAGUS runtime improved considerably as a result, but with negligible impact to accuracy. From my perspective, while these improvements are excellent from a codebase improvement perspective and are surely welcome to the MAGUS userbase, they seem to be fairly standard approaches and, in the context of the original MAGUS manuscript (which I have been reading side-by-side next to this manuscript), do not seem sufficiently novel beyond the previous MAGUS manuscript to justify publication in PLOS Computational Biology (rather, this article seems more appropriate for a technical blog post or similar).

Reviewer #2: The manuscript “Recursive MAGUS: scalable and accurate multiple sequence alignment” by Vladimir Smirnov describes an extension of the earlier MAGUS alignment package (which itself is an extension of the even earlier PASTA alignment approach). The fundamental idea behind MAGUS and PASTA is straightforward – 1) sequences to align are broken into groups and each group is aligned; 2) the subalignments are then merged. This basic approach and need for good methods of very large-scale multiple sequence alignments is clear – the PASTA method has been cited almost 250 times in the 6 years since it was published. Thus, the important question for this manuscript is: doe Recursive MAGUS represents an important and useful extension of these approaches.

I’ve gone over the results quite carefully and believe the program is both straightforward and potentially quite useful. There is clear evidence that alignments produced by MAGUS are quite good under all settings tested (Figures 3, 5, 6, and 7). Run-times also appeared to be reasonable in general. The options used to run programs are clear. I do have one big question is about the lossy compression (discussed on page 4). I would like to see a figure depicting the algorithm (i.e., a flowchart with examples showing the operations on data columns). That is the one part of the paper where, try as I might, I simply could not understand what is being done by the program.

Another more philosophical area that the author might consider discussing is the potential uses of very large alignments. This issue came to me when I was trying to understand the lossy compression issue. For phylogeny it might be better to estimate several alignments, each of which is relatively large but not so large as to require the lossy compression, and then estimate trees from each and combine the trees using a supertree approach. It is not clear to me whether very large alignments would have benefits for studies of molecular evolution. Again, the alignments could be broken down into subsets and analyzed separately. Note that by “very large” I mean large enough to have to invoke lossy compression – it is clear that large alignments are useful.

Finally, I would like to apologize for a delayed review. Some unexpected stresses on my time came up after I agreed to review. I kept thinking I’d get to the review but then got buried under other obligations. Please accept my sincere aplogy.

**Have the authors made all data and (if applicable) computational code underlying the findings in their manuscript fully available?**

Reviewer #1: Yes

Reviewer #2: Yes

PLOS authors have the option to publish the peer review history of their article (what does this mean?). If published, this will include your full peer review and any attached files.

Reviewer #1: **Yes: **Niema Moshiri

Reviewer #2: **Yes: **Edward L Braun
---

## [Decision Letter · Decision Letter 1]

9 Sep 2021

Dear Mr. Smirnov,

We are pleased to inform you that your manuscript 'Recursive MAGUS: scalable and accurate multiple sequence alignment' has been provisionally accepted for publication in PLOS Computational Biology.

Best regards,

Dina Schneidman

Software Editor

PLOS Computational Biology

Reviewer's Responses to Questions

**Comments to the Authors:**

Reviewer #1: With the expansion of the lossy compression scheme portions of the paper, I now feel as though it is sufficiently novel with respect to the original Smirnov & Warnow (2020) paper to justify its own publication. The lossy compression approach is quite clever, and it is much more clearly communicated with the additional diagram. Great work!

**Have the authors made all data and (if applicable) computational code underlying the findings in their manuscript fully available?**

Reviewer #1: Yes

PLOS authors have the option to publish the peer review history of their article (what does this mean?). If published, this will include your full peer review and any attached files.

Reviewer #1: **Yes: **Niema Moshiri

---

## [Editor Report · Acceptance letter]

28 Sep 2021

PCOMPBIOL-D-21-00627R1 

Recursive MAGUS: scalable and accurate multiple sequence alignment

Dear Dr Smirnov,

I am pleased to inform you that your manuscript has been formally accepted for publication in PLOS Computational Biology. Your manuscript is now with our production department and you will be notified of the publication date in due course.

With kind regards,

Katalin Szabo
